# Experimental manifestation of redox-conductivity in metal-organic frameworks and its implication for semiconductor/insulator switching

Jingguo Li[1,3], Amol Kumar [1,3], Ben A. Johnson [1,2] & Sascha Ott [1]✉

Electric conductivity in metal-organic frameworks (MOFs) follows either a band-like or a redox-hopping charge transport mechanism. While conductivity by the band-like mechanism is theoretically and experimentally well established, the field has struggled to experimentally demonstrate redox conductivity that is promoted by the electron hopping mechanism. Such redox conductivity is predicted to maximize at the mid-point potential of the redox-active units in the MOF, and decline rapidly when deviating from this situation. Herein, we present direct experimental evidence for redox conductivity in fluorine-doped tin oxide surface-grown thin films of Zn(pyrazol-NDI) (pyrazol-NDI = 1,4-bis[(3,5-dimethyl)-pyrazol-4-yl]naphthalenediimide). Following Nernstian behavior, the proportion of reduced and oxidized NDI linkers can be adjusted by the applied potential. Through a series of conductivity measurements, it is demonstrated that the MOF exhibits minimal electric resistance at the mid-point potentials of the NDI linker, and conductivity is enhanced by more than 10000-fold compared to that of either the neutral or completely reduced films. The generality of redox conductivity is demonstrated in MOFs with different linkers and secondary building units, and its implication for applications that require switching between insulating and semiconducting regimes is discussed.

Electroactive metal-organic frameworks (MOFs) have become fascinating research targets with potential applications in reconfigurable electronics, microporous conductors, energy conversion and storage, etc.[1–4]. However, a differentiated understanding of the underlying charge transport mechanisms within these low atomic density materials is still not fully achieved. To a certain extent, the interpretation of charge transport fundamentally relies on whether MOFs should be considered as conventional solids with continuous energy bands, or as a coordination polymers of redox-active units with strong electronic localization[5]. In the former case, electrical conductivity can be rationalized with the band theory of solid-state physics and, correspondingly, MOFs have been engineered with the aim of having greater band dispersion. Generally, this is achieved either by formation of effective delocalized electronic orbitals between linkers and metal nodes (through bond or extended conjugation) or by creation of strong π-π stacking effects between aromatic linkers (through space)[1]. Based on these principles, MOFs with different conductivity properties, ranging from semiconductors to metal analogues, have been obtained[6–8]. For

[1]Department of Chemistry–Ångström Laboratory, Uppsala University, 75120 Uppsala, Sweden. [2]Technical University of Munich (TUM), Campus Straubing for Biotechnology and Sustainability, Uferstraße 53, Straubing 94315, Germany. [3]These authors contributed equally: Jingguo Li, Amol Kumar. ✉e-mail: Sascha.ott@kemi.uu.se

example, 2D cobalt/iron 2,3,6,7,10,11-triphenylenehexathiolate MOFs exhibit metallic-like character at certain temperatures as a result of strong band dispersion where a fixed level of conductivity would be expected over the applied voltage (Fig. 1a, the representative MOF is shown in Fig. 1b)[9,10]. As the band dispersion decreases, a bandgap between the maximum of the valence band and the minimum of the conduction band emerges. In such cases, semiconductor characteristics take over, and the degree of conductivity is specified by the Fermi-Dirac statistics (Fig. 1c, the representative MOF is shown in Fig. 1d)[11].

In the absence of any band dispersion, charge transport can be achieved through electron-hopping between discrete molecular sites in different redox states[12–15]. The success of a single electron-hopping event is fundamentally determined by the probability that a neighboring acceptor site is available. Therefore, maximum redox conductivity is expected when the oxidized and reduced sites are in equal population, which is the case at the standard potential of the redox active unit according to the Nernst equation[16]. As the applied potential deviates from the standard potential, the presence of either the occupied or unoccupied sites approaches zero, and the redox conductivity decays quickly to negligible levels. Overall, a characteristic bell-shaped distribution of the redox conductivity as a function of the applied potential can be expected for materials that conduct exclusively through the redox-hopping mechanism (the analytical expression for the plot in Fig. 1e is included in Supplementary information). This charge transport mechanism has previously been theoretically predicted and experimentally demonstrated by Murray and coworkers in non-porous conductive polymers[17,18]. The bell-shaped redox conductivity has however never been experimentally demonstrated in a microporous MOF material despite the fact that many electroactive MOFs have been developed which specifically follow the redox-hopping pathway (a representative MOF is shown in Fig. 1f)[19–26]. Instead, macroscopic redox-hopping processes in transient experiments have been the primary focus, particularly in terms of comparing apparent electron diffusion coefficients ($D_e^{app}$). In this regard,

extensive knowledge has been accumulated by investigating macroscopic redox-hopping as a function of intrinsic properties like MOF anisotropicity[22], pore size[24], redox-active site content[23], etc., as well as extrinsic factors, for example, the nature of counterions and solvents[21,25,26]. The observation of the bell-shaped conductivity curve that is centered around the standard potential of the redox active unit has been hampered in MOFs that conduct by a mixed band conduction/redox hopping mechanism[27–30]. In addition, poor film quality with crystal boundaries or surface defects may further obscure the identification of redox conductivity.

In the present work, $Zn^{2+}$-pyrazolate-based MOF with redox-active naphthalene diimide groups, Zn(pyrazol-NDI) (pyrazol-NDI = 1,4-bis[(3,5-dimethyl)-pyrazol-4-yl]naphthalenediimide) was selected as a model MOF to investigate electron hopping charge transport and redox conductivity. This MOF consists of redox-innocent tetrahedral $d^{10}$ $Zn^{2+}$ ions and redox-active NDI linkers with molecular-like redox features due to strong electronic localization. Importantly, the coordination environment minimizes metal-ligand d-π orbital overlap and π–π electronic coupling between neighboring linkers, and makes electron-hopping the primary channel for charge transportation. In order to keep effects from grain boundaries to a minimum[31], Zn(pyrazol-NDI) was not immobilized as a bulk powder, but directly grown as a thin film on fluorine-doped tin oxide (FTO) electrode surface. The NDI linkers feature two sequential one-electron reductions, and we envisioned that when half of the NDI linkers are reduced to NDI[•−], or half of NDI[•−] to NDI[2−], the redox conductivity would maximize[16,17]. Herein, we report the experimental bell-shaped distribution of redox conductivity as a function of the redox state of the MOF. The latter is controlled by the applied electrochemical potential, and clearly demonstrated by UV/Vis spectroelectrochemical characterizations. The generality of redox conductivity in MOFs with electronically isolated redox-active sites is confirmed in a selection of other MOFs. We propose that the observation of such bell-shaped conductivity is essential to identify electroactive MOFs that follow a redox hopping

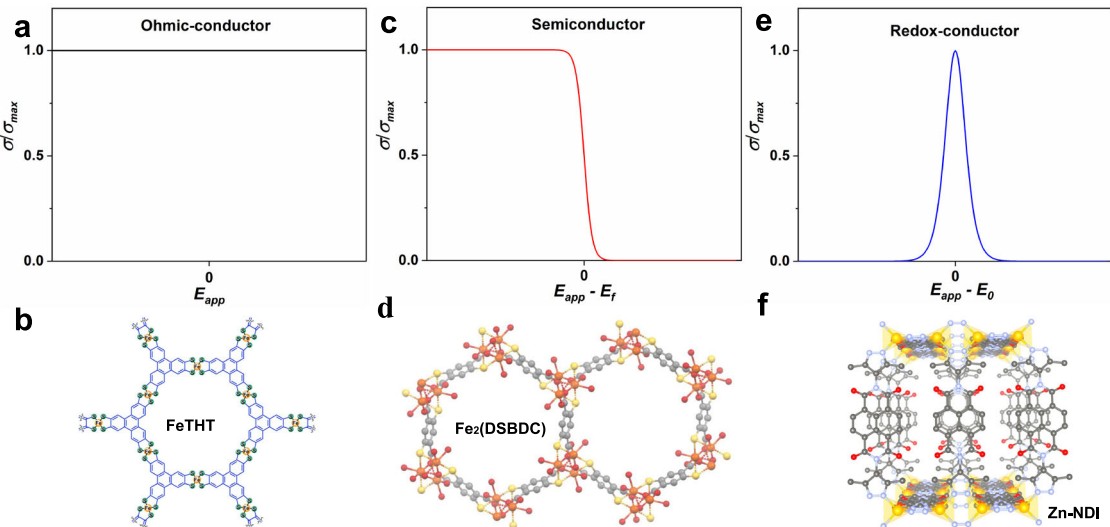

**Fig. 1 | Schematic diagrams of various metal organic framework (MOF) conductors and representative examples. a, b** Ohmic-conductor, where a fixed level of conductivity or resistance is associated so that a linear relationship is presented between the applied voltage and the current output (Ohms law, **a**); representative metallic conducting FeTHT (THT = 2,3,6,7,10,11-triphenylenehexathiolate) MOF with extensive electron delocalization throughout two-dimensional sheets (**b**, adapted with permission from ref. 9 Copyright 2019 American Chemical Society). **c, d** Semiconductor, where the magnitude of conductivity is specified by the Fermi-Dirac statistics. Here, the level of conductivity is presented at room temperature (**c**); representative semiconducting $Fe_2(DSBDC)$ (DSBDC = 2,5-disulfhydrylbenzene-1,4-dicarboxylic acid) MOF with band gap of 1.92 eV (**d**, picture is reproduced using the *Cif* file in ref. 11). **e, f** Redox conductor, where the probability of redox-hopping is determined by the occupancy of neighboring sites. It maximizes when the population of oxidized and reduced sites is equal (at the standard potential according to the Nernst equation, **e**); representative redox-conductive Zn-NDI (NDI is pyrazolate-functionalized redox-active naphthalene diimide-based linkers) MOF with linker-to-linker localized redox-hopping (**f**, reconstructed based on simulated model from ref. 19).

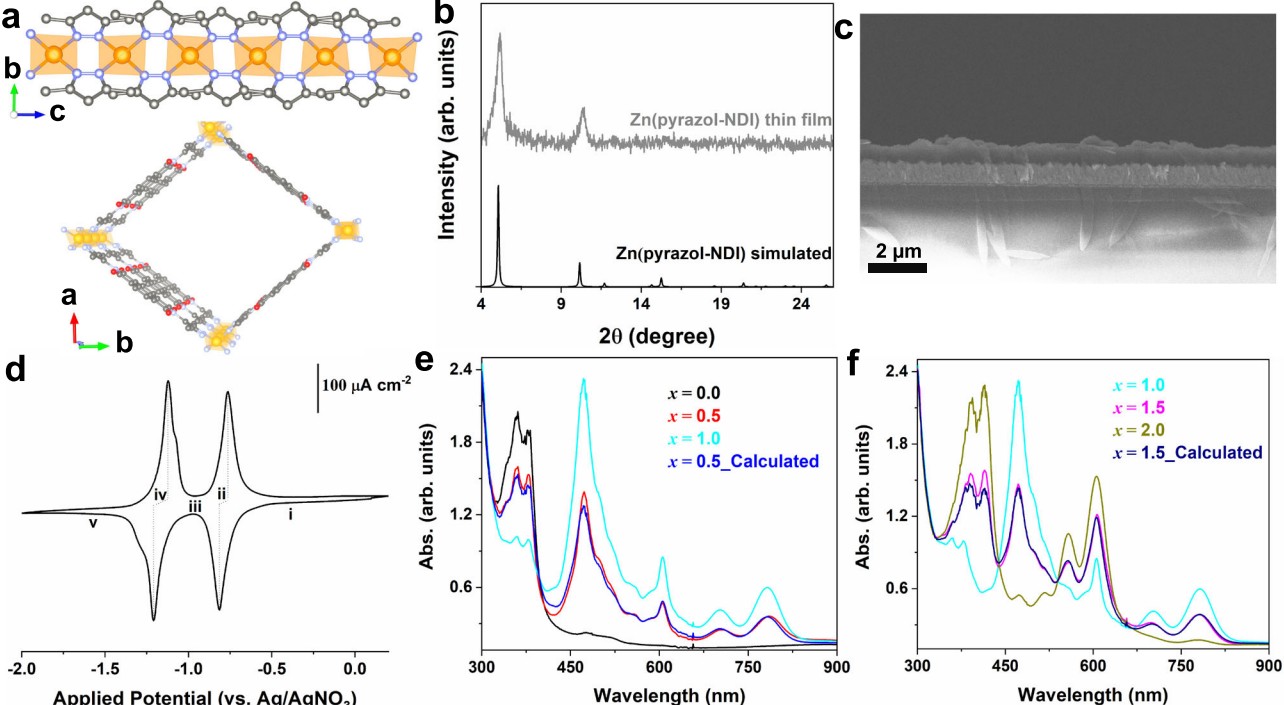

**Fig. 2 | Structural characterization, cyclic voltammetry (CV) and spectro-electrochemical measurements of Zn(pyrazol-NDI) thin-films. a** The crystal structure of Zn(pyrazol-NDI) with the projection of the pyrazolate-bridged $Zn^{2+}$ chain (top) and viewing along the c direction (bottom); hydrogen atoms are omitted for clarity; C, N, O, and Zn atoms or ions are presented with gray, blue, red, and brown spheres, respectively (the crystal structure was reconstructed based on the reported structure in ref. 19). **b** Experimental thin-film X-ray diffraction (XRD) data on fluorine-doped tin oxide surface (FTO) surfaces and simulated powder XRD data. **c** Scanning electron microscopy (SEM) cross-section image of the Zn(pyrazol-NDI) thin-film on FTO. **d** A representative CV with a scan rate of 5 mV s⁻¹: points i-v indicate various potentials, the applications of which result in different redox states of the film as described by the mole fraction of electrons (x, ranging from 0 to 2 with a step of 0.5) with respect to the NDI linkers. **e, f** UV−vis spectro-electrochemical measurements of Zn(pyrazol-NDI) thin-film at controlled potentials to access different mole fraction of electrons within the film, x-Zn(pyrazol-NDI), x = 0.0, 0.5, 1.0 (**e**) and x = 1.0, 1.5, 2.0 (**f**), calculated spectra are also included for x = 0.5, 1.5 using linear combination of x = 0.0 and 1.0, x = 1.0 and 2.0, respectively. All electrochemistry data were collected in Ar-saturated DMF with KPF₆ as the supporting electrolyte (0.1 M).

mechanism. As such, the present study will aid in identifying the correct primary conductivity mechanism, which is important for both fundamental and applied future studies.

## Results

### MOF thin-film preparation and basic characterizations

The Zn(pyrazol-NDI) MOF thin-film on FTO was prepared according to a previously reported procedure (see the "Methods" section for MOF synthesis, Supplementary Figs. 1–8)[20]. Structural modelling was performed to incorporate a distortion angle of 77° between two neighboring linkers connected to the same metal node, which makes the 1D channel oblique rather than square[19]. This structural feature is well-known in dipyrazolate-coordinated MOFs in which an infinite chain of $Zn^{2+}$ nodes are tetrahedrally coordinated by pyrazolate ligands (Fig. 2a top)[32,33]. The estimated effective NDI•••NDI distance is 8.03, 9.58, and 3.86 Å along the a, b, and c axis, respectively (Fig. 2a bottom), and thus feasible for redox-hopping[22]. Thin-film X-ray diffraction (XRD) experiments confirmed the preferred orientations of the MOF on the FTO surface with distinct (110) and (220) peaks (Fig. 2b). As a result, the c-axis of the material is parallel to the FTO surface. More importantly, the geometric arrangement of the neighboring NDI linkers precludes direct intermolecular π-orbital overlap, leaving redox-hopping rather than band transport as the primary charge transport pathway. Scanning electron microscopic (SEM) images further confirmed the compact and uniform thin-film growth (Fig. 2c and Supplementary Fig. 2). The thickness of the thin-film is estimated to be ~700 nm as suggested by cross-section characterizations (Fig. 2c), see Supplementary Fig. 3 for measurement details.

## Electrochemical modulation of the thin-film redox state and spectroelectrochemical identification

A fundamental requirement for the demonstration of redox conductivity in a MOF is the precise control of the its redox state[12,15]. In this study, the redox activity of the installed pyrazol-NDI linkers was characterized by cyclic voltammetry (CV). The diffusional electron hopping charge transfer gives rise to discrete waves for the NDI/ NDI⁻ and NDI⁻/NDI²⁻ couples. Specifically, a slow scan CV was conducted where the redox change in the thin-film framework is in a finite diffusion regime (Fig. 2d). In other words, the timescale of the CV experiment is slower than the timescale for diffusional electron hopping charge transport, resulting in the reduction of the entire film (scan rate-dependent CVs that show the transition to the semi-infinite diffusion regime are provided in Supplementary Figs. 5, 6)[34]. Two well-separated and reversible reduction waves are observed at −0.79 and −1.17 V vs Ag/AgNO₃. Both formal potentials are consistent with those of the homogeneous linker (Supplementary Fig. 7), as expected for MOFs in which the metal nodes and linkers are electronically isolated. In a standard Nernstian process, the redox state of the MOF thin-film can be expressed in terms of x-Zn(pyrazol-NDI), where x stands for the mole fraction of electrons, ranging from 0 to 2, depending on the applied potential and the resulting degree of NDI reduction (details about the notation are included in Supplementary information). Therefore, the selection of any specific redox state with compositional precision is possible, for instance, 0.5-Zn(pyrazol-NDI) (Fig. 2d-ii, reduction of half of the number of neutral NDI to NDI⁻) and 1.5-Zn(pyrazol-NDI) (Fig. 2d-iv, further reduction of half of the NDI⁻ to NDI²⁻) are accessible at applied potentials corresponding to the formal

potentials of NDI/NDI$^{\cdot-}$ (−0.79 V vs Ag/AgNO$_3$) and NDI$^{\cdot-}$/NDI$^{2-}$ (−1.17 V vs Ag/AgNO$_3$). We notice that such an electrochemical reduction strategy has been rarely explored for redox state modulation of MOFs[35]. Instead, chemical reductions or oxidations are frequently used to obtain the desired stoichiometries[27–30,36,37], which requires complementary experimental measurements to confirm the resulting redox state. In the present study, the electrochemical addressable redox states of the thin-film are confirmed by operando UV–Vis spectroelectrochemical measurements (details in the Methods section). The distinct electronic transitions are identical to those of the molecular species (Fig. 2e, f), and free from band-like absorbance[38]. The neutral thin-films (Fig. 2d, i) display characteristic π–π* excitations of the NDI core at 360 and 379 nm (Fig. 2e, $x = 0.0$)[26]. When the potential is held at −0.96 V vs Ag/AgNO$_3$ (Fig. 2d, iii), new excitations which are assigned to the NDI$^{\cdot-}$ species are observed at 473, 606, 703 and 782 nm (Fig. 2e, $x = 1.0$)[26]. Mathematically, a linear combination of $x = 0.0$ and $x = 1.0$ will give a half-reduced thin-film ($x_{calcd.} = 0.5$, Fig. 2e) which has the formulation of 0.5-Zn(pyrazol-NDI). Indeed, we observed an identical electronic excitation profile (Fig. 2e, $x = 0.5$) when the applied potential is held at the formal potential of the NDI/NDI$^{\cdot-}$ couple (−0.79 V vs Ag/AgNO$_3$). After the complete second reduction (Fig. 2d, v), a new pair of electronic transitions appears at 394 and 415 nm (Fig. 2f, $x = 2.0$), assigned to the formation of NDI$^{2-}$ [26]. Similarly, the calculated linear combination of $x = 1.0$ and $x = 2.0$ ($x_{calcd.} = 1.5$, Fig. 2f) matches the experimental results of electrochemically obtained $x = 1.5$ (Fig. 2f). Such stoichiometric quantification based on these spectroscopic analyses provide compelling evidence for not only the assignment of the redox states of the thin-films, but also the molecular nature of these electronic transitions, which is crucial for the Zn(pyrazol-NDI) to act as a redox conductor.

## Redox state dependent conductivity and semiconductor/insulator switchability

To probe the redox conductivity of MOFs at different redox states (as a function of $x$), thin-films were subject to different applied potentials for two minutes to equilibrate to the desired redox state (monitored by chronoamperometry; see Supplementary Fig. 8). Subsequently, and in analogy to literature precedence[22,39–41], electrochemical impedance spectroscopy (EIS) was used to probe the respective redox conductivities (details can be found in the Methods section). This AC voltammetry technique applies a small voltage perturbation over a wide range of frequencies, differentiating physical processes happening at distinct time scales. Importantly, this small perturbation approach promises a near-steady-state measurement (towards the intrinsic microscopic redox conductivity, σ) in which the concentrations of oxidized and reduced species change periodically in a constant cycle and net counterion ingress or egress can be neglected. To highlight the effect of the redox state of the film, Bode plots of experimental impedance data are presented as a function of applied potentials (Fig. 3a, the data points at the frequency of 0.1 Hz were magnified to facilitate comparison). Very interestingly, the evolution of the maximum impedance over the applied potential range follows the trend of the CV in Fig. 2d, with two minima appearing around the formal potentials of the NDI/NDI$^{\cdot-}$ and NDI$^{\cdot-}$/NDI$^{2-}$ couples. The change in impedance upon modulating the redox state can be better demonstrated by directly comparing experimental Nyquist plots of 0.0-Zn(pyrazol-NDI) and 0.5-Zn(pyrazol-NDI), or 1.5-Zn(pyrazol-NDI) and 2.0-Zn(pyrazol-NDI) (Supplementary Figs. 9, 10). Quantitative analyses of the corresponding redox conductivity with the equivalent circuit (EC) that considers purely electron transport (Supplementary Fig. 11a) show a bell-shaped distribution as highlighted in Fig. 3b. The experimental impedance data were further fitted with ECs that consider specifically a semi-infinite diffusion Warburg element (Supplementary Fig. 11c) and a constant phase element

(Supplementary Fig. 11b). Interestingly, a consistent evolution feature of the redox conductivity was observed as a function of external applied potential (or redox state) for all three ECs, each displaying two well-separated bell-shaped curves centered around the same potentials (see Supplementary Fig. 12 and a detailed discussion in the SI). In other words, irrespective of whether the experimental data is analyzed with ECs that consider electronic or ionic diffusion as the limiting process, qualitatively similar redox conductivity features, i.e., bell-shaped conductivity profiles are obtained, as one would expect from phenomena that arise from cation-coupled electron hopping transport[25].

Comparable redox conductivity (in the range of $10^{-10}$ to $10^{-9}$ S cm$^{-1}$) was observed when the thin-film is either in its neutral state (applied potential > −0.5 V vs Ag/AgNO$_3$, $x = 0.0$), or after the complete two-electron reduction (applied potential <−1.4 V vs Ag/AgNO$_3$, $x = 2.0$). The bell-shaped curves spread over $0.0 < x < 1.0$ and $1.0 < x < 2.0$ where the thin-film is in the mixed-redox states. The maximum redox conductivity is in the range of $10^{-5}$ to $10^{-6}$ S cm$^{-1}$ when the applied potential is equivalent to the formal potentials of the NDI/NDI$^{\cdot-}$ and NDI$^{\cdot-}$/NDI$^{2-}$ couples. Redox conductivity of the thin films is thus 10000-fold higher in the mixed-redox state ($x = 0.5$ or 1.5) compared to the situation when the films are in single oxidation states ($x = 0.0$, or 2.0). With such a difference in conductivity, the MOF thin-film lies between the regimes of insulators (∼$10^{-10}$ S cm$^{-1}$) and semiconductors (∼$10^{-6}$ S cm$^{-1}$), thus offering opportunities for applications in, for example, MOF-based field-effect transistors (FET)[42]. Such applications are further encouraged by the excellent insulator/semiconductor switchability between $x = 0.0$ and $x = 0.5$ over 100 cycles (Fig. 3d), with only minimum changes of the electrochemical features observed after a long-term stability test (Supplementary Fig. 13). In addition, morphological as well as structural stability of the MOF thin-film after electrochemical operation was confirmed by SEM and thin-film XRD characterizations (Supplementary Figs. 14–16). It is important to mention that the bell-shaped redox conductivity curve, as shown in Fig. 3b, is, to the best of our knowledge, the first of its kind reported for a MOF and is an intrinsic property for redox-conducting systems (as shown in Fig. 1e). This unique redox conductivity feature is fundamentally determined by the bimolecular nature of the electron self-exchange reaction, where it demands the coexistence of neighboring reduced and oxidized sites for successful redox conduction. With MOFs offering precise control over their chemical and structural properties, they provide opportunities to tune the redox conductivity channel at the nanoscale, and ultimately warrant a strong structure-function correlation between electronic state and redox conductivity. We also want to highlight that the diagnostic redox-conductivity feature may in some cases be convoluted by the interference with co-existing band conduction as observed in previous investigations[27–30,36,43].

To further understand the potential-dependent conductivity at steady-state, variable-temperature EIS measurements were conducted for the mono-valent (0.0-Zn(pyrazol-NDI)) and mixed-valent (0.5-Zn(pyrazol-NDI)) thin-films. As shown in Fig. 3c, the redox conductivity is thermally activated in both cases, a feature that correlates well with the nature of redox-hopping[41,44]. The experimental data was fitted with a Arrhenius-type relation ($\sigma T = \sigma_0 e^{-E_a/k_B T}$) to obtain the activation barriers[8]. This treatment results in formal numbers for the activation barriers of 82.9 meV (1.91 kcal mol$^{-1}$) for the 0.0-Zn(pyrazol-NDI (at −0.4 V vs Ag/AgNO$_3$), while that for 0.5-Zn(pyrazol-NDI is 40.4 meV (0.93 kcal mol$^{-1}$). While this twofold decrease parallels with the observed decrease in resistivity, it is inconsistent with the underlying theory (Supplementary Equation 1) which assumes purely electronic self-exchange and therefore identical activation energies. Non-ideal behaviors that arises from ion-pairing, intermolecular interactions, non-unity activity coefficients, etc. can however be expected for conductivity driven by ion-coupled electron hopping charge transport. These effects may vary depending on $x$, and the calculated activation

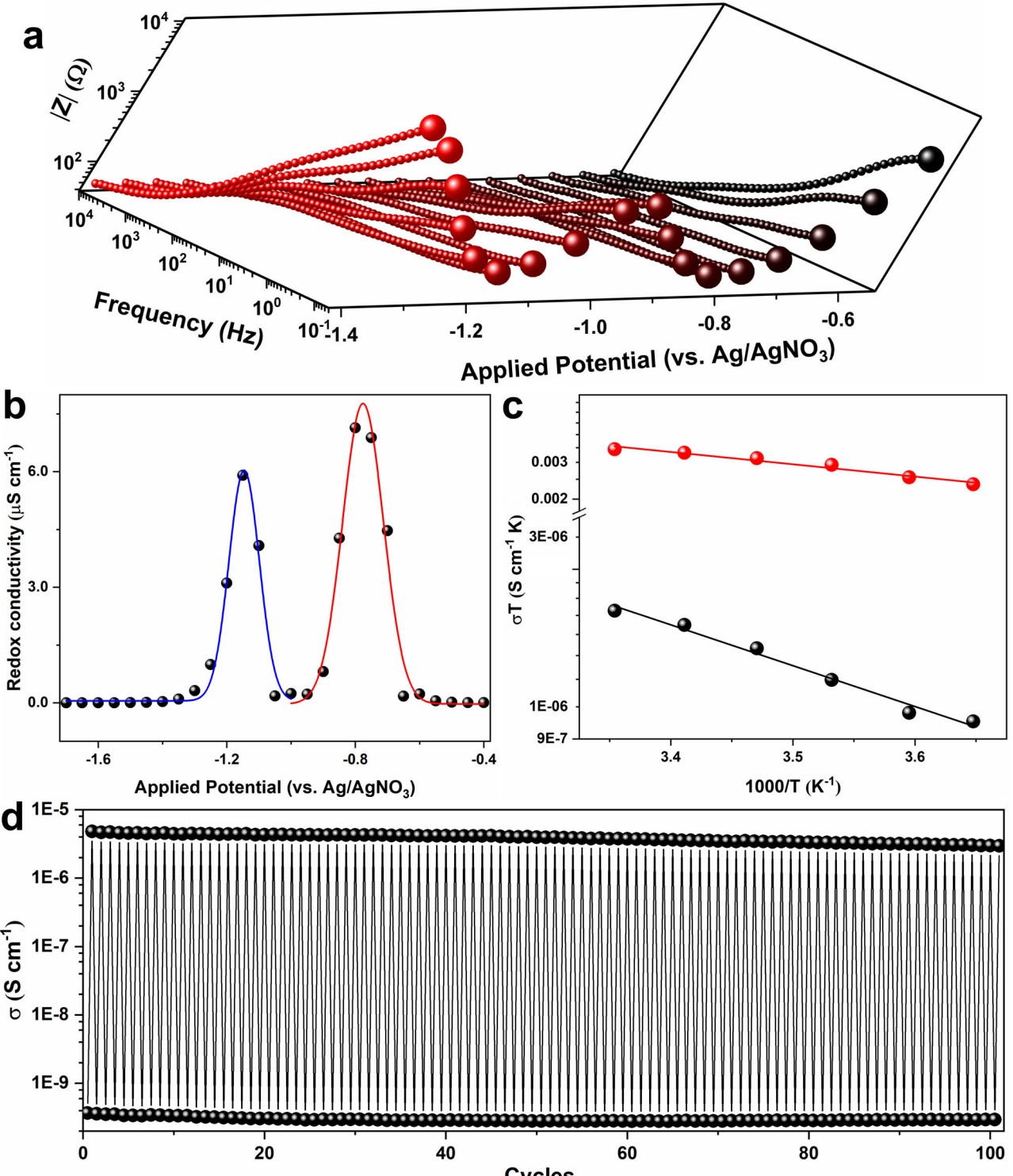

**Fig. 3 | Redox-conductivity characteristics of Zn(pyrazol-NDI) thin-films, together with activation barrier and conductivity switching measurements.** **a** Bode plots of Zn(pyrazol-NDI) thin-film at different applied potentials (the impedance data point at the frequency of 0.1 Hz was magnified for each measurements to highlight the effect of redox state); before each measurement, a stabilization time of 120 s was applied at respective potentials to achieve steady redox states. **b** Evolution of the steady-state thin-film conductivity as a function of applied electrochemical potential, which determines the mole fraction of electron reduction, $x$-Zn(pyrazol-NDI), $0.0 \leq x \leq 2.0$, raw data is available in the source data file. Gaussian fit was performed for NDI/NDI$^{\cdot-}$ based (red line) and NDI$^{\cdot-}$/NDI$^{2-}$ (blue line) bell-shaped redox conductivity. **c** Steady-state conductivity of 0.0-Zn(pyrazol-NDI) (black sphere) and 0.5-Zn(pyrazol-NDI) (red sphere) thin-films as the function of temperature, activation energy is derived based on the Arrhenius equation. **d** Switchability of the redox conductivity between 0.0-Zn(pyrazol-NDI) (bottom sphere) and 0.5-Zn(pyrazol-NDI) (top sphere) over 100 cycles (~24 h operation).

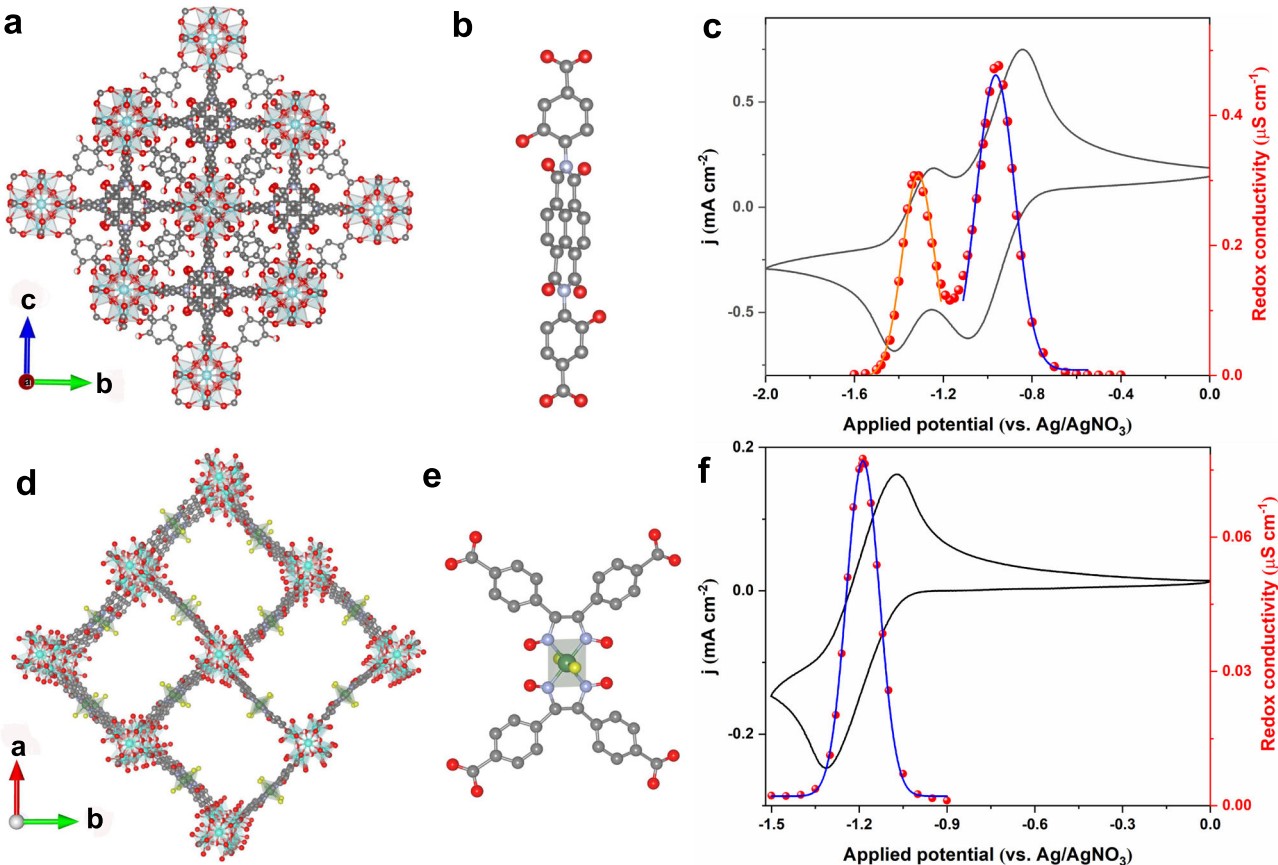

**Fig. 4 | Crystal structures, redox-active linkers, cyclic voltammetry (CV) and redox-conductivity of Zr(dcphOH-NDI) and UU-100(Co) thin-films. a** Crystal structure of Zr(dcphOH-NDI) viewing along a axis and displaying two inter-penetrated frameworks having a 12-c **fcu** net (the crystal structure was reconstructed based on the reported structure in ref. [25]). **b** Structure of the redox-active dcphOH-NDI linker. **c** Representative CVs (100 mV s$^{-1}$) and redox-conductivity characterizations of Zr(dcphOH-NDI) thin-films. Gaussian fit was performed for NDI/NDI$^-$ based (blue line) and NDI$^-$/NDI$^{2-}$ (brown line) bell-shaped redox conductivity. d Crystal structure of UU-100(Co) viewing along the c axis and displaying

a tetragonal unit cell ($a = b = 27.3$ Å, and $c = 19.6$ Å) with *P4/mbm* space group (the crystal structure was reconstructed based on the reported structure in ref. [45]). **e** Structure of the cobaloxime-based redox-active linker. **f** Representative CVs (100 mV s$^{-1}$) and redox-conductivity characterizations of UU-100 thin-films. Gaussian fit was performed for Co$^{2+}$/Co$^{1+}$ based (blue line) bell-shaped redox conductivity. All measurements were conducted in Ar-saturated DMF with KPF$_6$ as the supporting electrolyte (0.1 M). Hydrogen atoms are omitted for clarity; C, N, O, Co, Cl, and Zr atoms are presented with brown, light blue, red, yellow and green spheres, respectively. Redox conductivity data is available in the source data file.

---

energies should thus not be over-interpreted. In fact, large differences in the CVs when changing to different electrolytes point towards such effects, and will be presented below (Fig. 5).

## Generality of the redox conductivity with other MOFs

To validate the concept of redox conductivity that has been identified in Zn(pyrazol-NDI) MOF thin-films, two more electroactive MOF thin-films were prepared: (i) Zr(dcphOH-NDI) with an identical redox-active NDI core but different anchoring groups and secondary building units (SBUs) as shown in Fig. 4a, b [25,26], and (ii) UU-100(Co) with a cobaloxime-based redox-active linker [45] (Fig. 4d, e, more details about MOF synthesis and characterizations can be found in the "Method" section and Supplementary Figs. 17–21, 26–29). The linkers in both MOFs are electronically isolated redox active units, and feature as reversible CV waves (Fig. 4c, f). Moreover, film reduction can be monitored by UV–Vis spectroelectrochemical measurements, similar as in the case of Zn(pyrazol-NDI) [26,45]. Redox conductivity measurements with EIS and redox conductivity analysis were performed for thin-films of both MOFs (Supplementary Figs. 22–25, 30–32). Both materials consistently reproduce the characteristic bell-shaped curves of redox conductivity, with maximum conductivities at the respective formal potentials of −1.05 and −1.29 V for Zr(dcphOH-NDI) (Fig. 4c),

and −1.19 V for UU-100(Co) (Fig. 4f). This finding indicates that the characteristic bell-shaped redox conductivity curve should be expected as a universal phenomenon for electroactive MOFs that follow the redox conductivity mechanism, the identification of which will be important for future MOF-based electronics developments. Our results also highlight that a high-quality electroactive MOF thin-film with (i) good crystallinity, (ii) electronically isolated redox active units, and (iii) reversible electrochemical redox waves are prerequisites for the observation of intrinsic redox conductivity.

## Counter cation dependent redox conductivity

Electron hopping charge transport is intrinsically coupled to the translocation of charge-balancing counter cations [25,35,42]. It is thus interesting to probe how the above-described bell-shaped redox conduction curve is influenced by counter ions different from K$^+$. Zn(pyrazol-NDI) was investigated in the presence of Li$^+$ and tetrabutylammonium (TBA$^+$) with the former being known for strong ion-pairing effects, while the latter is considerably larger in size (experimental EIS data and redox conductivity analysis shown in Supplementary Figs. 33–36 for Li$^+$, Supplementary Figs. 37–40 for TBA$^+$) [25]. CVs and steady-state thin-film conductivity measurements indicate that both cations facilitate the first one-electron reduction of the NDI linkers (Fig. 5). The

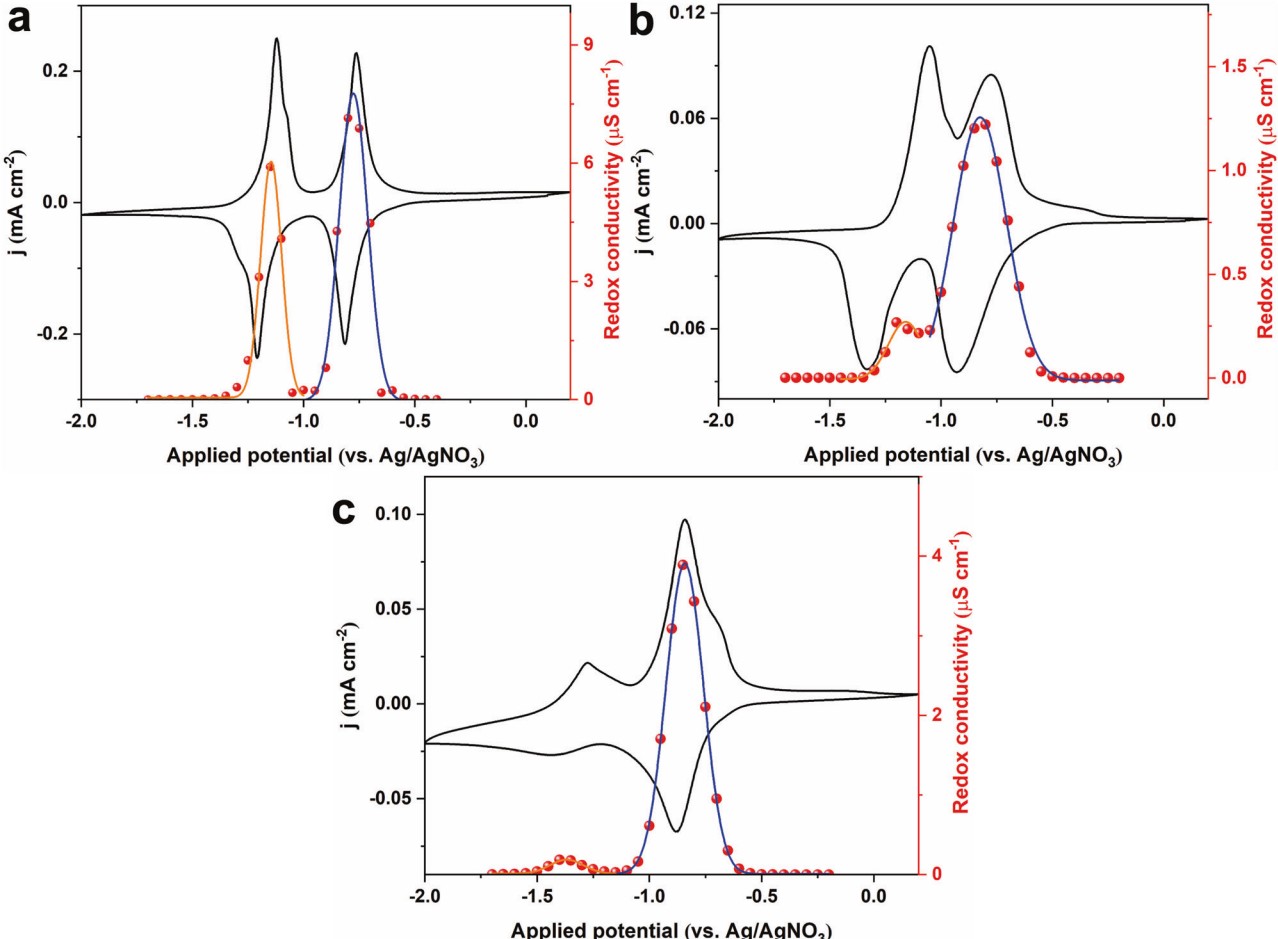

**Fig. 5 | Electrochemical and steady-state redox conductivity measurements with different counter cations.** Cyclic voltammetry (CV) scan (5 mV s$^{-1}$) and steady-state redox conductivity measurement of Zn(pyrazol-NDI) thin-films in Ar-saturated DMF with KPF$_6$ (**a**), LiClO$_4$ (**b**), and TBAPF$_6$ (**c**) as the supporting electrolyte (0.1 M). Gaussian fit was performed for NDI/NDI$^{-}$ based (blue line) and NDI$^{-}$/NDI$^{2-}$ (brown line) bell-shaped redox conductivity for all three different counter cations. Redox conductivity data is available in the source data file.

characteristic bell-shaped distribution of the redox conductivity was reproduced and centered near the respective formal potentials, highlighting the fact that charge transport in the Zn(pyrazol-NDI) MOF is following the redox conduction mechanism. However, the maximum redox conductivity for 0.5-Zn(pyrazol-NDI) has notably dropped from 7.1 µS cm$^{-1}$ (K$^+$ as counterions, Fig. 3b) to 3.9 µS cm$^{-1}$ (TBA$^+$ as counterions, Fig. 5c) and further to 1.2 µS cm$^{-1}$ (Li$^+$ as counterions, Fig. 5b, for direct comparison see Supplementary Fig. 41a). Variable-temperature conductivity measurements confirmed much higher formal activation barriers for 0.5-Zn(pyrazol-NDI) when using Li$^+$ (150 meV, Supplementary Fig. 42a) or TBA$^+$ (142 meV, Supplementary Fig. 42b) rather than K$^+$ (40.4 meV, Fig. 3c) as counterions. Both observations suggest that Li$^+$ and TBA$^+$ cations are experiencing strong ion-pairing and size effects, respectively, during the steady-state redox conduction, and the microscopic electron hopping is still cation-coupled, even though there is no net macroscopic ingress or egress[12]. This microscopic cation-coupled electron hopping is even more severe in the second reduction, where a significant redox peak separation and lower peak current density are clearly observed when Li$^+$ and TBA$^+$, respectively, were utilized as counterions (Fig. 5). In fact, this non-ideal electron hopping behavior is also evident when fitting the data with the underlying theory (Supplementary Equation 1). Nevertheless, the intrinsic redox conduction property is still identifiable, while the maximum conductivity is significantly reduced (Fig. 5 and Supplementary Fig. 41b).

## Discussion

Compared to the widely accepted band-like charge transport mechanism in MOFs, redox conductivity and its experimental manifestation has received much less attention. Herein, we present Zn(pyrazol-NDI) as an ideal system to showcase redox conductivity in MOFs. Zn(pyrazol-NDI) features minimal electron delocalization between metal nodes and organic linkers, as well as between linkers themselves. The electronic isolation of the redox-active units is consistent with the well-defined redox chemistry of the Zn(pyrazol-NDI), while corresponding UV–Vis spectroelectrochemistry studies allow monitoring of the redox states of the thin-film. We demonstrate experimentally the potential-dependent bell-shaped conductivity for a MOF-based redox conductor. The specific conductivity is highly dependent on the redox level of the thin-film and the nature of the counterions. The redox conduction mechanism was independently identified in two more MOF thin-films with different SBUs or different redox-active linkers, highlighting the general validity of redox conductivity as an underexplored conductivity mode in MOFs. Our findings clearly show that redox conductivity values in MOFs like the ones discussed herein must be reported together with the redox state of the material. The current work also proposes electrochemical impedance spectroscopy as a relatively easily performed method to differentiate between MOFs that should be treated with a band theory model, and those that are redox conductors. The excellent insulator-semiconductor switchability endows such thin-film MOFs promising

applications in the fabrication of micro- to nano-structured electronic devices for switchable memory elements. The results herein also challenge the conventional beliefs that MOFs that rely on diffusional charge transport are poor candidates for (multielectron)electro-catalysis, as most conductivity studies of such material have been performed at potentials at which the materials are insulating, neglecting the bell-shaped redox conductivity.

## Methods

### Synthesis of Zn(pyrazol-NDI) thin-film on FTO surface

The synthesis of the pyrazol-NDI linker follows previous protocols[19], and the purity of the linker was confirmed by [1]H NMR spectroscopy. FTO slides were cut into $1 \times 2$ cm plates and cleaned successively in solutions of DI water, ethanol, acetone by sonication for 20 min each. Cleaned FTO slides were placed in a 20 ml scintillation vial with the conductive side facing down. To this vial, a solution of the pyrazol-NDI linker (4.3 mM) and $Zn(NO_3)_2 \cdot 6H_2O$ (4.8 mM) in 10 ml DMF was added, and the sealed vial was placed into an oven, heated slowly from room temperature to 135 °C (heating rate 5 °C/min), and held at this temperature for 3 h. After that, the reaction mixture was slowly cooled to room temperature at a rate of 2 °C/min, and the obtained thin-films were washed with DMF. The unwanted growth on the glass side of the FTO was carefully removed by Kimtech wiping paper wetted with DMF. Subsequently, the cleaned thin-films were sonicated in DMF for 1 min to remove loosely bonded particles. Finally, the Zn(pyrazol-NDI) thin-films were thoroughly washed with DMF and stored in DMF for future use.

### Synthesis of Zr(dcphOH-NDI) thin-film on FTO surface

The synthesis of the dcphOH-NDI linker followed published protocols[25,26], and the purity of the linker was confirmed by [1]H NMR spectroscopy. Zr(dcphOH-NDI) thin-films on FTO surface were prepared similarly as previously described with minor modifications. Cleaned FTO ($1 \times 2$ cm) slides were immersed in the dcphOH-NDI linker solution (1 mM in anhydrous DMF) for 12 h to form a self-assembled monolayer (SAM). The SAM-modified FTO was then washed with DMF and kept in DMF while preparing solutions for MOF thin-film growth. In a separate 20 mL vial, dcphOH-NDI linker (53.8 mg, 0.1 mmol), $ZrCl_4$ (23.3 mg, 0.1 mmol), anhydrous acetic acid (286 μL, 5 mmol) and DMF (8 mL) were mixed and sonicated to form a homogeneous solution. The SAM-modified FTO was then placed in the precursor solution with the FTO side facing down. The vial was sealed and incubated in a sand bath for 3 days at 120 °C. After the solvothermal growth, the reaction was slowly cooled to room temperature, and the obtained thin-films were washed with DMF. The unwanted growth on the glass side of the FTO was carefully removed with a Kimtech wiping paper wetted with DMF. The prepared Zr(dcphOH-NDI) thin-film was soaked in DMF for future analysis.

### Synthesis of UU-100(Co) thin-film on FTO surface

The synthesis of the cobaloxime linker followed published protocols[45], and the purity of the linker was confirmed by [1]H NMR spectroscopy. UU-100(Co) thin-films on FTO were prepared similarly as previously described with minor modifications. Cleaned FTO ($1 \times 2$ cm) was immersed in the cobaloxime linker solution (1 mM in anhydrous DMF) for 12 h facing up to form a SAM. The SAM-modified FTO was then washed with DMF and kept in DMF while preparing solutions for MOF thin-film synthesis. In a separate 20 mL vial, $ZrCl_4$ (26 mg, 0.11 mmol), anhydrous acetic acid (35 μL, 0.61 mmol), 50 μL $H_2O$, and DMF (8 mL) were mixed and sonicated to form a homogeneous solution. The vial was placed in a pre-heated oven (80 °C) for 2 h to form the Zr cluster solution. After cooling naturally to room temperature, the cobaloxime linker (30 mg, 0.038 mmol) was added to the Zr cluster solution and sonicated to form a homogeneous solution for MOF synthesis. The SAM-modified FTO was then transferred into the precursor solution with the FTO side facing down. The vial was sealed and incubated in a sand bath for 5 days at 80 °C. After the solvothermal growth, the reaction mixture was slowly cooled to room temperature, and the obtained thin-films were washed with DMF. The unwanted growth on the glass side of the FTO was carefully removed with a Kimtech wiping paper wetted with DMF. The prepared UU-100(Co) thin-film was soaked in DMF for future analysis.

### Electrochemistry and UV–vis spectroelectrochemistry

Cyclic voltammetry (CV) and chronoamperometry was performed on a Metrohm Autolab potentiostat (PGSTAT302) with Nova 2.1.4 software in a one-compartment, three-electrode configuration. All measurements were performed in Ar-saturated DMF solutions using 0.1 M $KPF_6$, $LiClO_4$ or $TBAPF_6$ as the supporting electrolyte, with Zn(pyrazol-NDI) thin-film on FTO and a glassy carbon rod as working electrode and counter electrode respectively. A nonaqueous Ag/Ag($NO_3$) (0.01 M in acetonitrile) reference electrode was used (measured as $-0.09 \pm 0.02$ V vs. $Fc^{+/0}$). UV-vis spectroelectrochemistry was conducted by combining the above-mentioned electrochemistry setup with a Varian Cary 50 UV–vis spectrophotometer. A cuvette was employed as the electrochemical cell, and all other conditions remained the same as above. Absorption spectra were collected in kinetic mode along with the electrochemistry operations.

### Redox conductivity and activation energy measurements

The redox conductivity of the MOF thin-film was carried out using electrochemical impedance spectroscopy (EIS) under the same experimental electrochemistry conditions as stated above. Thin-films were preconditioned at desirable applied potentials for two minutes using chronoamperometry. After that, EIS measurements were performed with a 10 mV AC potential modulation in a frequency range of 0.1–10,000 Hz. All experimental data were fitted with different ECs, which consider specifically a semi-infinite diffusion related Warburg element, a constant phase element, or a simple RC circuit without the diffusional element (see Supplementary Fig. 11). The resulting inter-site redox-hopping resistances were obtained from fits to the RC circuit. This resistance was then used to calculate the respective redox conductivity at different applied potential using Ohm's law: $\sigma = \iota / RA$, where $\sigma$ is the redox conductivity, $\iota$ is the thickness of the MOF thin-film, $R$ is the resistance related to inter-site redox hopping, and $A$ is the measurement area of the film. For the variable-temperature EIS measurements, the electrochemistry cell was precooled or warmed to the targeted temperature using a chiller with isopropanol as the running liquid. Before each measurement, the cell was kept for at least three minutes to get an equilibrated temperature across the thin-film. To estimate the activation energy, the calculated temperature-dependent redox conductivity data were fitted to a Arrhenius-type relation[8].

## Data availability

Supplementary Information is available in the online version of the paper. It includes full details of the synthetic procedures, (spectro) electrochemical studies and impedance measurements, and other supporting characterization data. Figure 1e was produced from Supplementary Equation 1 in the Supplementary Information. The raw redox conductivity data were fitted with Gaussian. Source data are provided with this paper.

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

## Acknowledgements
Financial support for the work was provided by the Swedish Energy Agency (P42029-2), the Knut & Alice Wallenberg Foundation (KAW 2019.0071), and the Olle Engkvists Foundation (212-0147). We thank Dr. Andrew K. Inge from Stockholm University for his help with modelling the structure of Zn(pyrazol-NDI), and Dr. Souvik Roy from Lincoln University, UK, for providing cobaloxime linker.

## Author contributions
J.L. and A.K. contributed equally to this work. J.L. and S.O. planned and designed the experiments. A.K. synthesized the NDI ligand, J.L. and A.K. prepared the thin-film MOF on FTO. J.L. and A.K. performed the thin-film XRD, SEM, spectroelectrochemistry characterization. J.L. collected and, together with B.A.J. analyzed the electrochemistry and redox conductivity data. J.L., B.A.J. and S.O. prepared the manuscript and all authors have commented and given consent to this publication.

## Funding

## Competing interests
The authors declare no competing interests.
