## [Peer Review File · Nature Communications]

This manuscript has been previously reviewed at another journal that is not operating a transparent peer review scheme. This document only contains reviewer comments and rebuttal letters for versions considered at *Nature Communications*.

REVIEWERS' COMMENTS

Reviewer #1 (Remarks to the Author):

The manuscript by Ott and co-workers describes a particularly interesting example of 'redox-conductivity' with respect to a Zn-pyrazoleNDI MOF and demonstrates the broad applicability of their approach to other electroactive MOFs. The work appears to have been competently performed and the conclusions are sensible. I consider it to be an important contribution to the growing field of electroactive MOFs.

I agree with the comments of the original reviewers both in regard to the quality of the work and the criticism that there are many examples in the literature of conductivity arising from 'redox-hopping' charge transport. I am satisfied with both the responses of the authors and the changes made to the manuscript. I am pleased to recommend publication of this fine work.

Reviewer #2 (Remarks to the Author):

While the authors have stepped back from the previous controversial comments regarding a redox hopping mechanism of electron transport through MOF, the main points brought by the previous reviewers regarding the novelty and significant impact of the work have not been elevated by conversion to a communication. I agree with the previous reviewers that the work is sound. However, it is well-known that a redox hopping material would have the greatest conductivity at the redox potential - confirmation of that point does not rise to the level of Nature Communications. Indeed, the work of Carl Brozek demonstrates the effect in some regard by using impedance spectroscopy.

Reviewer #3 (Remarks to the Author):

The authors have nicely addressed all comments and I recommend the paper be accepted.

RESPONSE TO REVIEWERS' COMMENTS

Reviewer #1 (Remarks to the Author):

The manuscript by Ott and co-workers describes a particularly interesting example of 'redox-conductivity' with respect to a Zn-pyrazoleNDI MOF and demonstrates the broad applicability of their approach to other electroactive MOFs. The work appears to have been competently performed and the conclusions are sensible. I consider it to be an important contribution to the growing field of electroactive MOFs.

I agree with the comments of the original reviewers both in regard to the quality of the work and the criticism that there are many examples in the literature of conductivity arising from 'redox-hopping' charge transport. I am satisfied with both the responses of the authors and the changes made to the manuscript. I am pleased to recommend publication of this fine work.

Reply: Thank you.

Reviewer #2 (Remarks to the Author):

While the authors have stepped back from the previous controversial comments regarding a redox hopping mechanism of electron transport through MOF, the main points brought by the previous reviewers regarding the novelty and significant impact of the work have not been elevated by conversion to a communication. I agree with the previous reviewers that the work is sound. However, it is well-known that a redox hopping material would have the greatest conductivity at the redox potential - confirmation of that point does not rise to the level of Nature Communications. Indeed, the work of Carl Brozek demonstrates the effect in some regard by using impedance spectroscopy.

Reply: Reviewer 2 might be referring to Fig. 10 in ref. #40:

These data basically reproduce what we have shown schematically for a semiconducting MOF in Fig. 1c, and does not illustrate redox conductivity. While EIS was used to measure "AC conductivity" for the material, this experiment certainly in no way demonstrates that a "redox hopping material would have the greatest conductivity at the redox potential" as Reviewer 2 suggests. The MOF clearly does not even display

a redox hopping mechanism. In fact, Brozek correctly assigns the mechanism as band type transport for their material. Redox conductivity and its maximum at a "50:50 mixture" are only mentioned in the discussion with reference to Murray's papers. If this is the work the Reviewer is referring to, they are again conflating the concepts of ohmic conduction, which operates by migration, and redox-hopping, which is a diffusional process.

Reviewer #3 (Remarks to the Author):

The authors have nicely addressed all comments and I recommend the paper be accepted.

Reply: Thank you.